

# Impact of coexisting type 2 diabetes mellitus on the urinary microbiota of kidney stone patients

Xiang Li[1,*], Yifan Tang[2,*], Zhenyi Xu[2], Hao Lin[2], Shichao Wei[2], Jiayi Sheng[2], Lei Hu[2], Shiyu Wang[3], Yu Zhao[3], Zhi Li[3], Chaowei Fu[3], Yifeng Gu[4], Qun Wei[5], Fengping Liu[3], Ninghan Feng[2] and Weiguo Chen[1]

[1] Department of Urology, The First Affiliated Hospital of Soochow University, Suzhou, China
[2] Department of Urology, Affiliated Wuxi No. 2 Hospital, Nanjing Medical University, Wuxi, China
[3] Wuxi School of Medicine, Jiangnan University, Wuxi, China
[4] Collaborative Innovation Center for Diagnosis and Treatment of Infectious Diseases, State Key Laboratory for Diagnosis and Treatment of Infectious Diseases, School of Medicine, The First Affiliated Hospital, Zhejiang University, Hangzhou, China
[5] Department of Surgical Oncology, Institute of Clinical Medicine, Sir Run Run Shaw Hospital, School of Medicine, Zhejiang University, Hangzhou, China
[*] These authors contributed equally to this work.

Corresponding authors
Ninghan Feng, N.feng@njmu.edu.cn
Weiguo Chen, chenweiguo1971@suda.edu.cn

## ABSTRACT

**Objectives.** Type 2 diabetes mellitus (T2DM) commonly complicates kidney stone disease (KSD). Our objective is to investigate the variations in the urinary microbiota between individuals with KSD alone and those with KSD plus T2DM. This exploration could have implications for disease diagnosis and treatment strategies.

**Methods.** During lithotripsy, a ureterscope was employed, and 1 mL of urine was collected from the renal pelvis after bladder disinfection. Sequencing targeting the V3–V4 hypervariable region was performed using the 16S rRNA and Illumina Novaseq platform.

**Results.** The Shannon index showed a significant decrease in the KSD plus T2DM group compared to the KSD-only group (false discovery rate = 0.041). Principal Coordinate Analysis (PCoA) demonstrated a distinct bacterial community in the KSD plus T2DM group compared to the KSD-only group (false discovery rate = 0.027). The abundance of *Sphingomonas*, *Corynebacterium*, and *Lactobacillus* was significantly higher in the KSD plus T2DM group than in the KSD-only group (false discovery rate < 0.05). Furthermore, *Enhydrobacter*, *Chryseobacterium*, and *Allobaculum* were positively correlated with fasting blood glucose and HbA1c values ($P < 0.05$).

**Conclusions.** The urinary microbiota in the renal pelvis exhibits differences between patients with KSD plus T2DM and those with KSD alone. Further studies employing animal models are necessary to validate these distinctions, potentially paving the way for therapeutic developments based on the urinary microbiota.

## INTRODUCTION

Kidney stones are a common urinary system disorder that can lead to severe lower back pain, kidney hydronephrosis, decreased kidney function, urinary tract infections, and other discomforts or complications in patients. It can also recur frequently, significantly impacting the quality of life. Kidney stones are a global health issue, with an estimated prevalence of around 12% in adults worldwide (*Alelign & Petros, 2018*). In China, the prevalence is on the rise. *Wang et al. (2017)* conducted statistical analysis of disease data from 1990 to 2016 and found that the prevalence has shown a phased increase in the past three decades, with rates of 5.96% from 1991 to 2000, 8.86% from 2001 to 2010, and 10.63% from 2011 to the present.

The exact causes of kidney stones remain unclear, but research has suggested an association with environmental factors (*Sakhaee, Maalouf & Sinnott, 2012*), and the microbiota has been a closely examined environmental factor in recent years (*Phillips, 2009*). Multiple studies indicate that patients have disruptions in their gut microbiota. For example, *Stern et al. (2016)* reported a 3.4-fold increase in the *Bacteroides* in the gut of patients compared to healthy individuals, while the *Prevotella* was 2.8 times lower in patients than in healthy individuals. They also found that *Bacteroides* were an independent risk factor for stone formation (*Stern et al., 2016*). Ticinesi and colleagues also found differences in the gut microbiota of stone patients compared to healthy individuals, such as lower microbial diversity in patients and lower relative abundance of *Faecalibacterium*, *Enterobacter*, and *Dorea* genera. Additionally, certain bacteria related to oxalate degradation were decreased in relative abundance in patients (*Ticinesi et al., 2018*).

Urinary microbiota is another area of human microbiota research that has gained attention in recent years. In 2012, researchers shattered the traditional notion of a "sterile bladder" using expanded quantitative urine culture and high-throughput sequencing techniques (*Wolfe et al., 2012*). The research revealed that the bladder, like other parts of the body, harbors a microbiota, and its microbial structure is related to individual health status (*Pearce et al., 2014*; *Thomas-White et al., 2018*). Since the bladder has a urinary microbiota and is connected to the renal pelvis through the ureter, it is plausible that the renal pelvis may also have a urinary microbiota. Therefore, our research team collected renal pelvis urine samples in recent years after disinfecting the bladder and indeed confirmed the presence of a urinary microbiota in the renal pelvi (*Liu et al., 2020b*). Given that diabetes is a common complication in stone patients (*Nerli et al., 2015*), and that elevated urinary glucose and uric acid due to diabetes can alter the renal pelvis microbial growth environment (*Daudon et al., 2006*), our research team aims to investigate whether stone patients with coexisting diabetes have an impact on the diversity and structure of the renal pelvis urinary microbiota.

## METHODS AND MATERIALS

### Patients

The study received ethical approval from the Ethics Review Board of the Second Peoples' Hospital of Wuxi (No: 201802) and was conducted between October 2018 and April 2019. All patients have informed consent and signed the subject informed consent form.

### Inclusion criteria for kidney stone disease (KSD) patients

Patients who had been diagnosed with calcium stones through X-ray, ultrasound, or CT scans and were willing to participate in the demographic survey.

### Exclusion criteria for KSD patients

Patients who were pregnant, menstruating, diagnosed with malignant tumors, autoimmune diseases, urethritis, prostatitis, benign prostatic hyperplasia (*Tsai et al., 2022*), renal cysts, bladder inflammation, urinary tract abnormalities, had undergone urinary catheterization within the past 4 weeks, or had used antibiotics or probiotic products within the past 4 weeks.

### Inclusion and exclusion criteria for patients with KSD and diabetes (KSD + DM)

In addition to meeting the above inclusion and exclusion criteria, these patients also had type 2 diabetes. The diagnostic criteria for diabetes were fasting blood sugar $\geq 7.0$ mmol/L or 2-hour postprandial blood sugar $\geq 11.0$ mmol/L (*Alberti, Zimmet & Consultation, 1998*).

All patients in the KSD cohort underwent ureteroscopic lithotripsy, a procedure during which stones were collected. The composition of the stones was determined using Infrared Spectrum analysis (Quest Diagnostics Inc., Secaucus, NJ, USA).

### Urine sample collection

The urine specimen collection process was previously outlined in our study (*Liu et al., 2020a*). In brief, a catheter is initially inserted through the cystoscope to obtain urine from the bladder. Subsequently, three successive rinses of the bladder are performed using iodine tincture. Finally, the bladder is thoroughly washed with saline solution until the withdrawn fluid becomes clear. Following bladder disinfection, a catheter is introduced through the ureteroscope to procure 1 mL of renal pelvis urine for bacterial DNA extraction.

### Bacterial isolation and bioinformatics

Data were collected as previously described in *Tsai et al. (2022)*. Specifically, the bacterial DNA isolation, 16 S rRNA sequencing, and bioinformatics (*Liu et al., 2023*). The extraction of bacterial DNA from urine, high-throughput sequencing, and bioinformatics analysis have been comprehensively described in our previous research (*Liu et al., 2022*). Here, we provide a brief overview: Following PCR amplification of bacterial DNA, extraction was carried out using AMpure XP magnetic beads (Beckman Coulter, Indianapolis, IN, USA). When conducting a microbiome study, it is essential to account for potential sources of contamination and validate the accuracy of the sequencing method. In our study, we included negative control and positive samples to assess environmental DNA

contamination (*Liu et al., 2022*). Amplification was performed using primers 319F and 806R, targeting the highly sensitive V3–V4 region of the 16S rRNA gene. Sequencing was conducted on the Illumina Novaseq platform.

Environmental contaminants of urine samples were conservatively removed as previously described. Briefly, bacterial ASVs whose counts did not exceed five times the maximum number of counts in the negative controls were considered as contaminates and removed, as described (*Liu et al., 2022*). We also manually removed bacteria that have been reported to be environmental contaminants from soil and water (*Liu et al., 2022*).

QIIME software was employed for sequence analysis, including quality adjustments, demultiplexing, and taxonomic assignment. Operational taxonomic units were determined using PiCRUSt based on the Greengenes database. Finally, diversity was assessed using QIIME, with distance calculations based on 97% similarity and unweighted UniFrac.

This project utilized R Studio (version 8.14; Altamor Drive, Los Angeles) for analysis. Specifically, Principal Coordinates Analysis (PCoA) was employed to assess the differences in microbial structures between the two groups. Taxonomy relative abundances were logarithmically transformed using Log2, followed by the application of the Wilcoxon test to determine inter-group differences in bacteria. Additionally, Benjamin-Hochberg correction was applied to adjust the p-values, where a corrected $P$-value ($P < 0.05$) indicated statistical significance.

## Statistical analysis

For continuous variables that conform to a normal distribution, we applied the $t$-test. In cases where they did not adhere to a normal distribution, the Wilcoxon test was utilized with Benjamin-Hochberg correction. A comparison of the differences in genera with an abundance higher than 0.5% between groups were performed. Categorical variables were subjected to the Pearson Chi-square test, and correlation analysis between two variables was performed using Pearson's analysis. Statistical significance was determined at a significance level of $P < 0.05$.

## RESULTS

### A comparison between KSD+DM and KSD-only groups

Table 1 presents a summary of the participants' demographic and clinical data. Here, age, BMI, HbA1c, FBG, glomerular filtration rate, blood urea nitrogen, blood uric acid, and blood creatinine were continuous variables followed a normal distribution, while stone duration and urine white blood cells were continuous variables which are not followed a normal distribution. Out of the 30 participants, 28 stones were identified as comprising 80% CaOx-monohydrate and 20% CaOx-dihydrate.

Both the KSD+DM and KSD-only groups exhibited an equal distribution of gender, as well as similar smoking and drinking habits ($P > 0.05$). No significant differences were observed between the two groups in terms of age, BMI, stone duration, glomerular filtration rate, urine white blood cell count, nitrite positivity in urine, or leukocyte esterase positivity ($P > 0.05$). As anticipated, HbA1c and fasting blood glucose levels were significantly higher in the KSD+DM group compared to the KSD-only group ($P < 0.05$). Furthermore,

**Table 1 A comparison between KSD+DM and KSD-only groups.** Demographics and clinical data of participants.

| Variables | KSD + DM ($n = 30$) | KSD ($n = 30$) | $t/Z/\chi^2$ | $P$-value |
|---|---|---|---|---|
| Female ($n$%) | 15 (50.00%) | 15 (50.00%) | 0.000 | 1.000 |
| Age (years) | 56.20 ± 11.41 | 55.00 ± 11.69 | 0.373 | 0.713 |
| BMI (kg/m$^2$ ) | 25.32 ± 2.06 | 24.65 ± 3.41 | 0.168 | 0.258 |
| Smoking ($n$%) | 3 (10.00%) | 3 (10.00%) | 0.000 | 1.000 |
| Drinking ($n$%) | 3 (10.00%) | 3 (10.00%) | 0.000 | 1.000 |
| HbA1c (%) | 7.66 ± 0.94 | 6.14 ± 0.94 | 1.922 | 0.009 |
| Fasting blood glucose (mmol/L) | 8.88 ± 2.65 | 5.75 ± 1.19 | 3.830 | 0.001 |
| Stone duration (days) | 274.35 ± 550.42 | 393.30 ± 1145.72 | −0.666 | 0.512 |
| Comorbid condition | | | | |
| Hypertension ($n$%) | 10 (33.33) | 8 (26.67) | 0.317 | 0.573 |
| Dyslipidemia ($n$%) | 11 (36.67) | 14 (46.67) | 0.617 | 0.432 |
| Glomerular filtration rate (ml/min/1.73m2) | 83.20 ± 13.23 | 98.10 ± 18.54 | −1.931 | 0.056 |
| Urine white blood cells (/ul) | 42.27 ± 39.10 | 111.33 ± 123.31 | −1.223 | 0.002 |
| Nitrite positive in urine ($n$%) | 0 (0.00%) | 0 (0.00%) | / | / |
| Leukocyte esterase positive ($n$%) | 12 (40.00%) | 18 (60.00%) | 0.800 | 0.371 |
| Blood urea nitrogen (mmol/L) | 6.22 ± 1.62 | 4.92 ± 1.32 | 2.006 | 0.060 |
| Blood uric acid (umol/L) | 374.92 ± 87.13 | 270.25 ± 61.62 | 3.102 | 0.006 |
| Blood creatinine (umol/L) | 78.82 ± 21.01 | 66.13 ± 25.14 | 1.181 | 0.233 |

**Notes.**

For continuous variables that followed a normal distribution, we used the $t$-test. If they did not adhere to a normal distribution, we employed the Wilcoxon test. For categorical variables, we applied the Pearson Chi-square test.

blood uric acid levels were notably elevated in the KSD+DM group in comparison to the KSD-only group ($P < 0.05$).

## The bacterial diversity exhibited differences between KSD+DM and KSD-only patients

As depicted in Fig. 1A, there was no statistically significant difference in the microbial richness index Chao 1 between the KSD+DM and KSD-only groups (false discovery rate >0.05); however, Fig. 1B demonstrated that the microbial diversity was notably higher in the KSD+DM group compared to the KSD-only group (false discovery rate = 0.041). Figure 1C demonstrates that the PCoA results showed a statistically significant $R^2$ value of 8% for distinguishing microbial structures between the groups (false discovery rate = 0.027). Due to hypertension and dyslipidemia being common comorbidities with kidney stone disease (KSD) and type 2 diabetes mellitus (DM), we analyzed whether these two comorbidities act as confounding variables in the urinary microbiota. The results revealed that both of them are not confounding factors ($P > 0.05$; Fig. S1). Figure 1D reveals that a total of 1,595 operational taxonomic units were detected in the urine of both groups. In the KSD+DM group, 67.68% of these operational taxonomic units were shared with the

KSD-only group, whereas the KSD-only group had 60.87% shared operational taxonomic units.

## Comparison of bacterial composition between groups

The top 10 bacterial genera in terms of relative abundance in this study included Acinetobacter, Aureimonas, Bacillus, Bifidobacterium, Delftia, Planococcaceae incertae sedis, Propionibacterium, Pseudomonas, Sphingomonas, and Staphylococcus (as shown in Fig. 2A). The bacterial genera that exhibited significant differences between the two groups included Sphingomonas, Corynebacterium, and Lactobacillus ($P < 0.05$; as shown in Fig. 2B).

## Correlation of fasting blood glucose and HbA1c values with differentially abundant bacteria between groups

The results of the correlation analysis between the relative abundance of 11 differentially abundant bacteria and fasting blood glucose and HbA1c values in the KSD+DM group revealed that the relative abundance of *Enhydrobacter*, *Chryseobacterium*, and *Allobaculum* genera exhibited correlations with fasting blood glucose and HbA1c values ($P < 0.05$; as shown in Table 2).

## DISCUSSION

The coexistence of kidney stones and diabetes leads to significant changes in the diversity and structure of urinary microbiota. Our findings indicate that individuals with both conditions exhibit higher diversity in their urinary microbiota compared to those with kidney stones alone. In a previous study conducted by our research team, we analyzed the urinary microbiota of 70 female patients with type 2 diabetes and compared it to 70 age- and gender-matched healthy individuals using midstream urine samples. Our results revealed a similar trend: the presence of diabetes was associated with increased microbial diversity (*Liu et al., 2017*). However, it's worth noting that this conclusion is not universally consistent. *Chen et al. (2019)* and colleagues conducted a study comparing the urine of 32 diabetes patients with 26 healthy individuals and found that while the abundance of bacteria decreased in diabetes patients, the overall diversity remained unchanged.

Furthermore, our study uncovered differences in the structural composition of urinary microbiota between the two patient groups. This finding aligns with similar observations reported by Chen and colleagues in their study on diabetes patients (*Chen et al., 2019*). In our previous research, although we observed visual differences in microbiota composition between the two groups, we did not achieve statistical significance (*Liu et al., 2017*). Additionally, a case-control study conducted by *Penckofer et al. (2020)* did not report such differences. Given that diet and ethnicity have been shown to influence gut microbiota (*David et al., 2014*; *Gaulke & Sharpton, 2018*), it is crucial to conduct comparisons within the same population to determine whether diabetes indeed leads to changes in urinary microbiota. Moreover, it's important to note that our study used renal pelvis urine samples, while studies by Chen J, Penckofer S, and our earlier research utilized clean midstream urine samples (*Chen, Zhao & Vitetta, 2019*; *Liu et al., 2017*; *Penckofer et al., 2020*). Alan J.

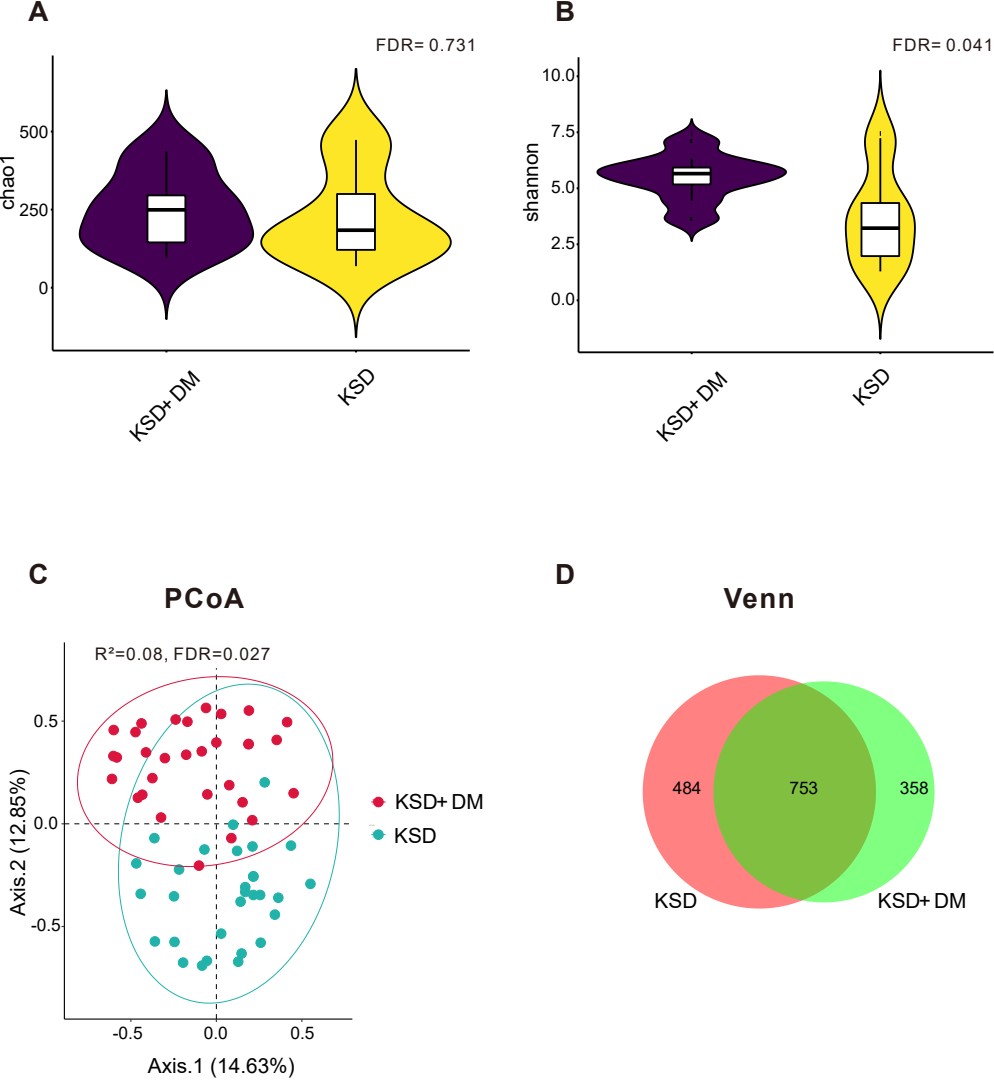

**Figure 1** **The bacterial diversity exhibited differences between KSD+DM and KSD-only patients.** Comparative analysis of bacterial microbiotas between the KSD+DM and KSD-only groups. (A) Bacterial richness was assessed using the Chao 1 index at the operational taxonomic unit level. The Wilcoxon rank-sum test was employed, and $P$-values were adjusted using the Benjamini and Hochberg false discovery rate. (B) Bacterial diversity was assessed using the Shannon index at the operational taxonomic unit level. The Wilcoxon rank-sum test was employed, and $P$-values were adjusted using the Benjamini and Hochberg false discovery rate. (C) Principal coordinate analysis (PCoA) based on Bray-Curtis distances at the operational taxonomic unit level revealed distinct microbial compositions between groups. The 95% confidence ellipse is depicted for each group. Statistical comparisons between the two groups were conducted using Permutational Multivariate Analysis of Variance (PERMANOVA). The $P$-value was adjusted using the Benjamini and Hochberg false discovery rate. (D) Venn diagrams were generated to compare ASV compositions among different groups.

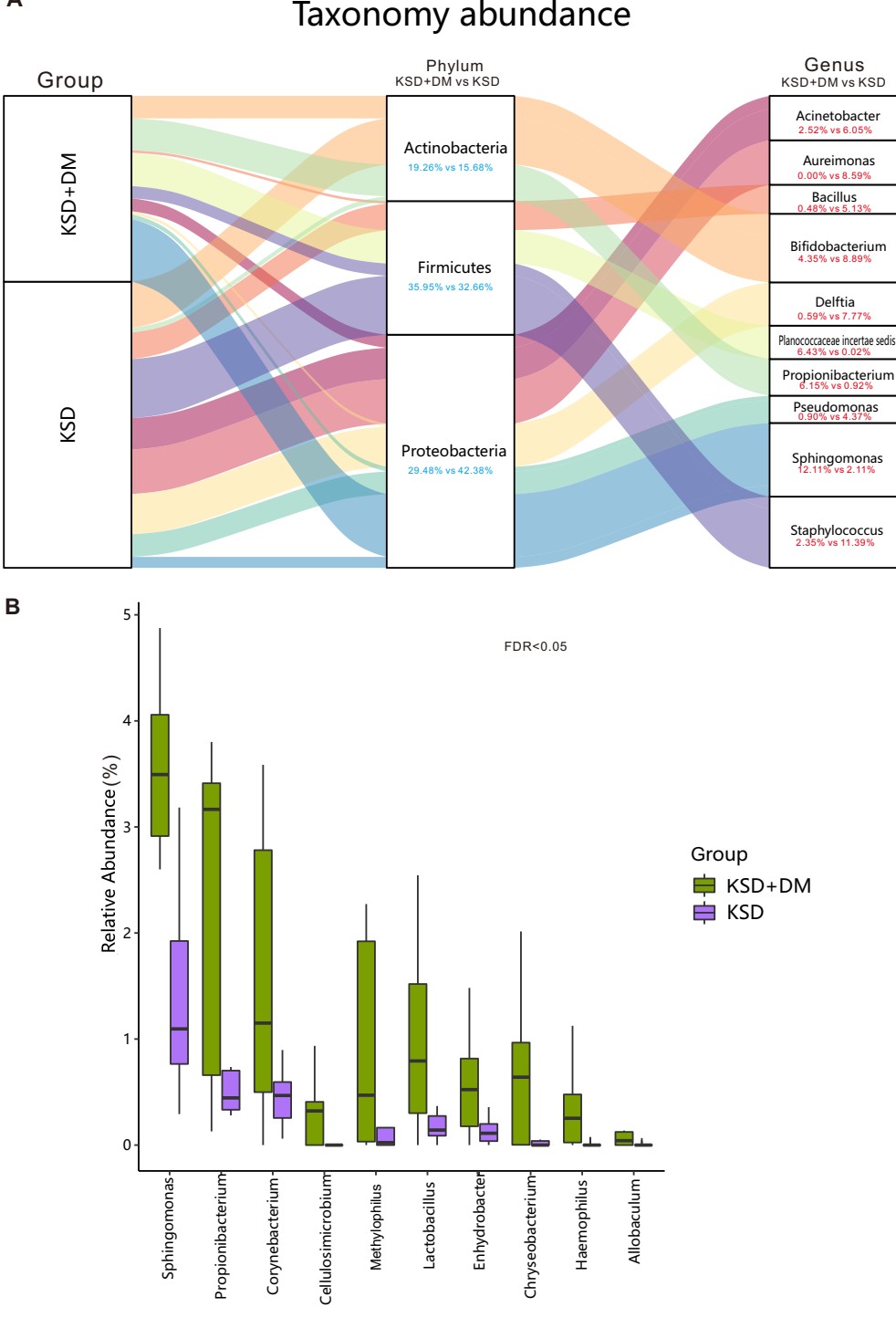

**Figure 2** **Comparison of bacterial composition between groups.** Bacterial profile in participants. (A) The ten most abundant bacterial genera, along with their corresponding phyla. (B) The genera displaying significant differences between groups have been presented. These differences were assessed using the Wilcoxon rank-sum test and adjusted for the Benjamini and Hochberg false discovery rate.

**Table 2 Correlation of fasting blood glucose and HbA1c values with differentially abundant bacteria between groups.**

| Parameters | Bacteria (KSD+DM vs KSD) | r | P-value |
|---|---|---|---|
| Fasting blood glucose | *Allobaculum (0.12 vs 0.00)* | 0.927 | <0.001 |
| | *Chryseobacterium (0.74 vs 0.19)* | 0.834 | 0.005 |
| | *Enhydrobacter (0.86 vs 0.12)* | 0.784 | 0.012 |
| HbA1c | *Allobaculum (0.12 vs 0.00)* | 0.893 | 0.001 |
| | *Chryseobacterium (0.74 vs 0.19)* | 0.817 | 0.007 |
| | *Enhydrobacter (0.86 vs 0.12)* | 0.864 | 0.003 |

Notes.
Pearson correlation analysis was performed. The data in parentheses represent the relative abundance of bacteria that show inter-group differences (KSD+DM vs KSD).

Wolfe and his colleagues have confirmed that clean midstream urine does not accurately represent the microbiota of the bladder or renal pelvis; it primarily reflects the "urethral microbiota" (*Brubaker et al., 2021*; *Wolfe & Brubaker, 2019*; *Wolfe et al., 2012*). Therefore, future research efforts should employ consistent methods for collecting renal pelvis or bladder urine samples to ensure the comparability of results across different studies.

The taxonomic hierarchy of urinary microbiota in patients with both kidney stones and diabetes has been altered. In our study, both the group with kidney stones only and the group with both conditions exhibited Proteobacteria as the dominant phylum in their urine, followed by Firmicutes and Actinobacteria. Interestingly, this finding contrasts with Penckofer S's study on the urinary microbiota of diabetes patients, where Proteobacteria were not the dominant phylum (*Penckofer et al., 2020*). In our earlier midstream urine study, we found that Proteobacteria were dominant in both diabetes patients and healthy individuals, ranging from 51.63% to 58.01% (*Liu et al., 2017*). In a study of the US population conducted by Pearce M M, Firmicutes were the primary phylum in bladder urine microbiota, followed by Actinobacteria and Proteobacteria (*Pearce et al., 2014*). This suggests that ethnicity plays a role in the distribution of bacterial phyla in urinary microbiota. However, further research comparing populations from different countries is required to confirm this observation.

In this study, it was observed that the relative abundance of Proteobacteria in patients coexisting with kidney stones and diabetes was slightly lower than in those without diabetes. In the context of gut microbiota, an increase in Proteobacteria has been recognized as one of the characteristics associated with metabolic disorders (*Shin, Whon & Bae, 2015*). Both kidney stones and diabetes are linked to metabolic disruptions. Therefore, the question arises: why do Proteobacteria, a phylum known to include various harmful bacteria, decrease when these two metabolic-related diseases coexist? To gain a clearer understanding of the reasons behind this phenomenon, further confirmation through large-scale studies is necessary.

In the earlier studies on urinary microbiota, the distribution of Proteobacteria remains unclear. For instance, Jiang et al. found that Proteobacteria was the most abundant phylum in three groups of patients, including those with kidney stone disease (KSD), patients with urinary tract tumors, and healthy controls (*Liu et al., 2017*). *Yang et al. (2023)* reported

an increase in Proteobacteria in patients with diabetic kidney disease compared to the controls. Additionally, *Pederzoli et al. (2020)* observed an elevation of Proteobacteria in patients with bladder cancer. Therefore, the role of Proteobacteria in urinary microbiota cannot be generalized, and its distribution may vary among different diseases. Future research, particularly large-sample multicenter studies, is needed to further explore this aspect.

Subsequently, functional experiments should be conducted to elucidate the role of Proteobacteria in urinary microbiota.

Notably, our study identified a significant increase in *Lactobacillus* in the urine of patients with both kidney stones and diabetes, a phenomenon also observed in previous studies on diabetes patients. In *Penckofer et al. (2020)*, the detection rate of *Lactobacillus* in diabetes patients' urine was higher than that in healthy individuals, and its relative abundance was positively correlated with HbA1c levels (*Liu et al., 2017*). Our previous study similarly found higher relative abundance of *Lactobacillus* in the urine of diabetes patients compared to healthy individuals. Like *Penckofer et al. (2020)*, our earlier research revealed that the relative abundance of *Lactobacillus* in diabetes patients' urine increased with rising fasting blood glucose levels (*Liu et al., 2017*). However, in our current study, we did not observe a correlation between the relative abundance of *Lactobacillus* in the urine of patients with both kidney stones and diabetes and diabetes diagnostic indicators such as fasting blood glucose or HbA1c. This discrepancy may be attributed to the relatively small sample size in our current study.

Furthermore, our study found a positive correlation between the presence of *Allobaculum* in the urine of patients with both kidney stones and diabetes and their fasting blood glucose and HbA1c levels. *Allobaculum* plays a probiotic role in the gut microbiota, contributing to anti-inflammatory responses, mucosal barrier protection, metabolic regulation, and immunomodulation (*Ma et al., 2020*). Therefore, if the role of bladder bacteria mirrors that of gut bacteria, the correlation between Allobaculum and fasting blood glucose and HbA1c may reflect the body's self-defense mechanisms. However, we also observed an increase in the harmful bacterium *Chryseobacterium* with rising fasting blood glucose and HbA1c levels. Although *Chryseobacterium spp.* has been linked to urinary tract infections and septicemia (*Acosta-Ochoa et al., 2013*; *Cascio et al., 2005*), its role in the microbial community is not well-documented. Thus, future research should consider conducting animal experiments to validate these findings.

The bacteria discovered in this study, such as *Sphingomonas*, *Propionibacterium*, and *Corynebacterium*, have been reported in several previous studies on urinary microbiota (*Ahn et al., 2022*; *Cappelli et al., 2023*; *Nickel et al., 2022*; *Perovic et al., 2022*; *Popovic et al., 2018*; *Kim & Park, 2018*). However, *Methylophilus*, identified in this study, has only been reported in earlier studies on human skin and gut microbiota (*Dekio et al., 2005*; *Jiang et al., 2021b*; *Jiang et al., 2021b*; *Lee et al., 2021*; *Zheng et al., 2023*). It is necessary to conduct large-scale studies in the future to further clarify the bacterial composition in human urine, which will play a crucial role in redefining urinary tract infections.

### Limitations to consider when interpreting our findings

One limitation of our study is the relatively small sample size. While our results offer valuable insights, larger cohorts could provide a more comprehensive understanding of the urinary microbiota in patients with kidney stones and diabetes. Future research with more extensive participant groups may help validate and strengthen our findings. Another consideration is the cross-sectional design of our study, which captures a snapshot of the urinary microbiota at a single point in time. Longitudinal studies that track changes over time would offer a more dynamic perspective on the interactions between kidney stones, diabetes, and the urinary microbiota. Such studies could reveal how these factors evolve and influence each other over extended periods. These limitations underscore the need for further investigation and the cautious interpretation of our findings. In addition, this study did not conduct animal model experiments to validate the mechanisms of urinary microbiota, which is also a limitation of this study. The main reasons for not conducting experiments on animal models to validate the mechanisms are as follows: (a) It is challenging to avoid damaging the renal tubules when transplanting urinary microbial communities into the kidneys or renal tubules of animal models; (b) the difficulty of extracting human urinary microbiota and transplanting it into the kidneys or bladder of animal models. This is because a significant proportion of bacteria in the human bladder are anaerobic, and we have not found a method for complete anaerobic collection and extraction of bladder microbial communities.

Recent studies have highlighted variations in the urinary microbiota in patients with kidney stones and diabetes compared to healthy individuals (*Chen et al., 2019*; *Liu et al., 2020b*; *Penckofer et al., 2020*; *Xie et al., 2020*). Given that both conditions can impact renal function and alter urine composition, this study is the first to explore differences in urinary microbiota in patients with both conditions compared to those with kidney stones alone. Additionally, certain bacteria were found to be correlated with diagnostic indicators of diabetes in these patients. To further confirm these findings and elucidate the causal relationship between the diseases and urinary microbiota, future research should prioritize larger sample size studies and conduct animal experiments.

## CONCLUSION

In summary, our study underscores the importance of considering urinary microbiota in kidney pelvis in the context of kidney stones and diabetes. The intricate interplay between these conditions and the urinary microbiota opens up new avenues for research and potentially novel approaches to managing and treating these health issues. As we continue to unravel the complexities of the human microbiome, future studies are poised to provide valuable insights into the pathophysiology of kidney stones, diabetes, and related metabolic disorders.

## ACKNOWLEDGEMENTS

The authors would like to express our gratitude to the study participants.

### Funding

This work was supported by the National Natural Science Foundation of China (82370777, 81874142 and 82073041), the Zhejiang Provincial Natural Science Foundation of China (LXR22H160001, and LY22H160011), and the Gusu Medical Talent Foundation (GSWS2020021). The funders had no role in study design, data collection and analysis, decision to publish, or preparation of the manuscript.

### Grant Disclosures

The following grant information was disclosed by the authors:
National Natural Science Foundation of China: 82370777, 81874142, 82073041.
Zhejiang Provincial Natural Science Foundation of China: LXR22H160001, LY22H160011.
Gusu Medical Talent Foundation: GSWS2020021.

### Competing Interests

The authors declare there are no competing interests.

### Author Contributions

- Xiang Li conceived and designed the experiments, performed the experiments, prepared figures and/or tables, authored or reviewed drafts of the article, and approved the final draft.
- Yifan Tang performed the experiments, authored or reviewed drafts of the article, and approved the final draft.
- Zhenyi Xu performed the experiments, authored or reviewed drafts of the article, and approved the final draft.
- Hao Lin performed the experiments, authored or reviewed drafts of the article, and approved the final draft.
- Shichao Wei performed the experiments, analyzed the data, authored or reviewed drafts of the article, and approved the final draft.
- Jiayi Sheng performed the experiments, authored or reviewed drafts of the article, and approved the final draft.
- Lei Hu performed the experiments, authored or reviewed drafts of the article, and approved the final draft.
- Shiyu Wang performed the experiments, prepared figures and/or tables, and approved the final draft.
- Yu Zhao performed the experiments, prepared figures and/or tables, and approved the final draft.
- Zhi Li performed the experiments, prepared figures and/or tables, and approved the final draft.
- Chaowei Fu performed the experiments, prepared figures and/or tables, and approved the final draft.
- Yifeng Gu performed the experiments, prepared figures and/or tables, and approved the final draft.

- Qun Wei performed the experiments, prepared figures and/or tables, and approved the final draft.
- Fengping Liu conceived and designed the experiments, performed the experiments, analyzed the data, prepared figures and/or tables, authored or reviewed drafts of the article, and approved the final draft.
- Ninghan Feng conceived and designed the experiments, prepared figures and/or tables, authored or reviewed drafts of the article, and approved the final draft.
- Weiguo Chen conceived and designed the experiments, prepared figures and/or tables, authored or reviewed drafts of the article, and approved the final draft.

## Human Ethics

The following information was supplied relating to ethical approvals (i.e., approving body and any reference numbers):

Ethics Review Board of the Second Peoples' Hospital of Wuxi (No: 201802)

## Data Availability

The raw sequencing data for this project are available at the GenBank Sequence Read Archive: PRJNA561017, SRP218817.

## Supplemental Information

Supplemental information for this article can be found online at http://dx.doi.org/10.7717/peerj.16920#supplemental-information.

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
