# Peer review of "Impact of coexisting type 2 diabetes mellitus on the urinary microbiota of kidney stone patients"

_PeerJ, doi:10.7717/peerj.16920_

## Round 0.1 · original submission · Major Revisions

Both reviewers have raised significant concerns that should be addressed with a revised manuscript. Please make sure to address all concerns, which, for some, can be addressed with a limitations paragraph in the Discussion, and modify the conclusions to be limited to what is supported by the evidence.

·

Basic reporting

This study explores the differences in urinary microbiota between individuals with kidney stone disease (KSD) alone and those with KSD and type 2 diabetes mellitus (T2DM). The methodology involves collecting urine during lithotripsy and utilizing sequencing techniques to analyze bacterial communities. The results indicate significant variations in bacterial diversity and composition between the two groups, suggesting distinct microbial profiles in the presence of T2DM. Specific bacterial abundances were highlighted, showcasing differences correlated with glucose and HbA1c values. The conclusion emphasizes these observed disparities in urinary microbiota, advocating for further validation through animal studies. This validation could pave the way for potential therapeutic advancements rooted in understanding these microbial differences. I have some comments as following:

Abstract
1. The abbreviation "FBG" should be represented as the full term, "fasting blood glucose.

Introduction
1. Should there have been a space in the first sentence on line 57?
2. On line 60, typically, in abbreviations, spaces are removed. So, " Wang W. Y.” or “Wang WY” would be the correct abbreviation for "Wang W Y.”

Methods and Materials
1. Why the authors exclude subjects with benign prostate hyperplasia?
2. On line 97, the authors refer to “that the urine specimen collection process was previously outlined in our study”. Is there a citation or reference supporting this as a standard urine sample collection method?
3. On line 103, the authors mention, "The extraction of bacterial DNA from urine, high-throughput sequencing, and bioinformatics analysis have been comprehensively described in our previous research." Is there a citation or reference supporting this statement?
4. On line 112, R Studio (version 8.14) should also indicate the company's location.
5. On line 118, could the authors specify which continuous variables followed a normal distribution and which did not?

Results
1. In Table 1, the number of category variables should not include decimal points.
2. In Table 1, the HbA1c levels did not show a significant difference between the two groups (p = 0.091). Could this be a typing error?
3. In Table 1, are the data for stone duration and urine white blood cells accurate? The same question applies to their P-values.
4. Table 2 should provide the full terms for FBG (fasting blood glucose).
5. In Figure 1, the first abbreviation occurred should use the full term, like FDR, OTU, and ASV.
6. In Figure 1, there appears to be a significant error in the description of Figure 1C, which depicts differences between “fecal and tissue groups.” However, this paper focuses on urine samples.

Discussion
1. On line 133, the authors state that microbial diversity was notably higher in the KSD+DM group compared to the KSD-only group (FDR = 0.041). However, on line 152, it's mentioned that individuals with both conditions exhibit lower diversity in their urinary microbiota compared to those with kidney stones alone. Which statement is accurate?
2. Although this paper focuses on DM and KSD, is it possible that subjects also have hypertension and dyslipidemia, common comorbidities with DM that might influence the results?
3. Do you have data on the stone composition, which could also potentially affect the results?
4. I haven't noticed significant hypotheses or key mechanisms explaining the differences in urine bacteria species between the KSD+DM and KSD groups. Could the authors elaborate further on the underlying mechanisms?
5. Between lines 182 to 189, could the authors provide a brief overview of existing theories or research on the potential role of Proteobacteria in urinary microbiota to enhance the depth of the discussion?
6. Please review the manuscript for grammar and check for any typos.

Experimental design

All my comments are listed in the basic reporting area.

Validity of the findings

All my comments are listed in the basic reporting area.

Additional comments

All my comments are listed in the basic reporting area.

Reviewer 2 ·

Basic reporting

The article is well-written and professionally done.

Experimental design

The experimental design is deeply flawed. The urine is known to be a low biomass niche of the body and therefor contamination is a major concern. The word 'contamination' does not appear in the manuscript.

Validity of the findings

The validity is in question because contamination is not addressed.

Additional comments

This article compares the microbiome of the renal pelvis between patients with kidney stone disease and those with kidney stone disease and type II diabetes. Overall, the manuscript is well-written and the analysis is appropriate, but the lack of any discussion regarding possible contamination of the samples is a fatal flaw.

In addition, I have several other concerns:

Figure 1- the y-axis font is too small and should be enlarged to be more legible for the reader.

Figure 1c- the authors mention fecal and tissue groups in the legend for figure 1, which obviously don't occur in this manuscript.

Line 134- the authors discuss figure 1b, but mean figure 1c.

Line 136- the authors discuss figure 1c, but mean figure 1d

Figure 2- what do the colors mean, if anything?

Figure 2b- All of the relative abundances are really low, with most below 1%. Is such a small difference really biologically significant? The authors would need to do more to make this convincing.

Table 2- What were the relative abundances for these genera? Why wasn't multiple testing correction done?

Line 155- The authors state, "Our results revealed a similar trend: the presence of diabetes was associated with reduced microbial diversity." This is the opposite of what they show in figure 1b.

The authors find some odd bacteria, such as Methylophilius, and others. Have other studies found these genera? More explanation is needed for genera that are not well known in human health.

---

## Round 0.2 · accepted · Accept

The authors have addressed the reviewers' comments, including Reviewer 2, who did not accept the invitation to review the revised manuscript.

·

Basic reporting

I have no other questions.

Experimental design

I have no other questions.

Validity of the findings

I have no other questions.

Additional comments

I have no other questions.